# Polycarbonate/Titania Composites Incorporating TiO_2_ with Different Nanoscale Morphologies for Enhanced Environmental Stress Cracking Resistance in Dioctyl Phthalate

**DOI:** 10.3390/polym14173693

**Published:** 2022-09-05

**Authors:** Yasir Khalid, Amine Achour, Muhammad Aftab Akram, Mohammad Islam

**Affiliations:** 1Australian Laboratory Services, 3-8 South Street, Sydney, NSW 2116, Australia; 2Pixium Vision S.A. 74 Rue du FGB Saint-Antoine, 75012 Paris, France; 3Department of Materials Science & Engineering, Pak-Austria Fachhoschule Institute of Applied Sciences & Technology, Khanpur Rd., Mang, Haripur 22620, Pakistan; 4GE Aviation Systems, 3290 Patterson Ave SE, Grand Rapids, MI 49512, USA

**Keywords:** polycarbonate, titania, composites, nanosheets, environmental stress cracking, hydrothermal process

## Abstract

Polycarbonate (PC) is susceptible to environmental stress cracking (ESC) when the conditions of pre-strain and presence of fluid with a compatible solubility index are both prevalent. One approach to counter this involves using nanoscale fillers to bridge the propagating microcracks, thus, effectively inhibiting impending failure. In this work, we report incorporation of titania (TiO_2_) with different nanoscale morphologies into polycarbonate matrix to assess its effect on ESC resistance against dioctyl phthalate (DOP). Using a hydrothermal process with a NaOH/Ti molar ratio of 72, TiO_2_ nanostructures were produced containing nanosheets with large surface area and nanotubes having typical diameter and length values of 15–20 nm and a few hundred nanometers, respectively. PC/TiO_2_ composites were fabricated with up to 0.5 weight percent of TiO_2_ nanoparticles (NPs), nanowires (NWs), or hybrid nanostructures (HNs). ESC tests were conducted by exposing test coupons to DOP oil at different temperatures and pre-strain conditions. The results showed that, under identical test conditions, while as-received PC grade exhibited complete fracture in ~3.1 h, PC/TiO_2_-0.05HN composite took ~70 h to fail via surface cracking. SEM examination of the fracture surface revealed that homogeneous dispersion and efficient load-bearing capability of TiO_2_ nanotubes and nanosheets impeded localized crack propagation by bridging the gap between the PC matrix segments. Liquid nitrogen fracture of the PC/TiO_2_ composite further confirmed the critical role of TiO_2_ hybrid nanostructures towards improvement in ESC resistance of PC matrix composites.

## 1. Introduction

Polycarbonate (PC) is a completely recyclable polymer and a favorable substitute to PVC in medical devices due to its ease of processing and superior mechanical properties. PC has also experienced increased consumption in the automotive OEMs and electrical/electronics segment. The global PC market size was USD 21.8 billion in 2021 and is projected to reach USD 35.4 billion by 2030 [1]. This tremendous growth is driven by key PC properties such as high ductility, rigidity up to 140 °C, toughness down to −20 °C, tunable mechanical properties, high thermal stability, and excellent dimensional stability. Due to its optical clarity, excellent flame resistance, high impact strength, and high stability in different environmental conditions, PC has found widespread use in the automotive, transportation, building and construction, packaging, medical, optical media and data-storage industries.

Environmental stress cracking (ESC) is a serious issue in PC when certain organic liquids cause cracking or crazing upon application of very modest stress, often an order of magnitude less than the actual tensile or flexural strength. The consequence of an ESC failure may range from merely cosmetic to life-threatening and has enormous industrial and economic repercussions. ESC involves crack generation and propagation on the surface and is also enhanced by the presence of residual stresses in the polymeric surface, presumably introduced during manufacturing and not eliminated during polymer annealing. In other words, the synergistic effects of chemical agents and the mechanical stresses cause polymer chain scission due to polymer degradation, thus, creating microscopic defects giving rise to surface initiation and rapid propagation [2,3,4,5]. The numerous environments encountered in application end-use include organic liquids (solvents), synthetic oils, naturally occurring oils, detergents/surfactants, mold release agents, plasticizers, sealants/caulking agents, adhesive formulations, inks, anti-corrosion additives, lubricants, and metal cutting fluids.

Ester plasticizers are commonly used in several thermoplastic and thermoset elastomer polymers since they offer low temperature or depression of glass transition temperature (T_g_) of various polymers without any significant loss of mechanical strength. Di-2-ethylhexyl phthalate or di-octyl phthalate, abbreviated as DOP, falls into the category of high molecular weight o-phthalate esters with low polarity and solubility parameter values. The molecular structure and properties of DOP are given in Figure 1. Ester plasticizers tend to dissolve or dilute amorphous or semi-crystalline polymers, and therefore, both the chemical structure and solubility parameter values of the polymer and the solvent must be kept in mind while considering plasticizers for ease of polymer processing or designing polymeric components for certain in-service conditions. For instance, if a molded polycarbonate-based medical device is in contact with flexible PVC containing some monomeric ester as a plasticizer, it is anticipated that the monomeric ester will migrate to the stressed polycarbonate surface, thus, inducing crazing and cracking. Another example could be of an over-torqued polycarbonate household appliance, with an electrical power housing or battery box connected with PVC sheath wire cables that may undergo wide atmospheric temperature variations.

Conventional approaches that have been explored to ameliorate ESC in polymers involve blending, particulate or fiber reinforcement, and impact modification. While no or limited improvement in ESC resistance has been reported in the case of particulate reinforcement, smaller fiber length and low ability to bridge the cracks developing from environmental exposure also restrict the effectiveness of fiber reinforcement [6,7,8,9]. With the advent of nanotechnology, fillers with nanoscale morphologies (particles, rods or tubes, and sheets) have gained attraction as a potential remedial approach towards ESC with the associated advantage of a low loading requirement due to high specific surface area and ability to modify surface chemistry for strong matrix/reinforcement interface. However, issues such as inhomogeneous dispersion and agglomeration among these fillers drastically limit their beneficial roles. There have been some reports on the effectiveness of SiO_2_ or silicone coating added to polycarbonate with limited or conditional success [2,10,11]. Most studies aimed at investigating the ESC issue in polycarbonate have covered at least one of the following aspects: (i) the type of environment (mild/moderate/harsh), (ii) blending with other polymers, and (iii) reinforcement with fillers (conventional/nanoscale).

Titania (TiO_2_) is an oxide that has found widespread use in many applications due to its existence as different polymorphs (anatase, Rutile, and Brookite), wide bandgap, and strong photocatalytic behavior. It can be synthesized with different nanoscale morphologies including nanoparticles, nanotubes, nanowires, and nanosheets [12,13,14]. The applications of this versatile material include pigments, sunscreens, supercapacitors, electrochemical electrodes, solar cells, and gas sensors [15,16]. In the present study, we explore the effects of nanoscale TiO_2_ morphology on the thermal properties and ESC resistance of the resulting composite in DOP oil. While the TiO_2_ nanoparticles (NP) and nanowires (NW) were procured from suppliers, TiO_2_ hybrid nanostructures (nanotubes and nanosheets) were synthesized using a hydrothermal process. After initial compounding of the PC granules with TiO_2_ NP, NW, or HN through melt blending, the molded plaque samples were thoroughly characterized for chemical composition and thermal properties. ESC tests in DOP with pre-defined constant strain were performed and the main source of improvement in the ESC resistance was determined from extensive electron microscopy of the fracture surfaces. This work highlights the key role of TiO_2_ hybrid nanostructures towards a significant improvement in ESC resistance.

## 2. Experimental Procedures

### 2.1. Materials

The PC grade used for this research was of opaque color with medium viscosity and UV stability (Lexan 163R, Sabic Industries). Titanium dioxide (TiO_2_) powders in the form of anatase phase nanoparticles (<25 nm, Sigma CAS#1317-70-0) and nanowires (100 nm diameter, 10 µm long, Sigma CAS# 13463-67-7) were procured.

TiO_2_ hybrid nanostructures consisting of nanosheets and nanotubes were synthesized using a hydrothermal process. Briefly, sodium hydroxide (NaOH) solution was made by adding NaOH pellets to water in a stainless steel reactor with a Teflon lining. A separate aqueous suspension of TiO_2_ nanoparticles was made with continuous magnetic stirring and ultrasonic vibration treatment. The molarity of the NaOH solution was maintained at 10 M with a NaOH/Ti molar ratio of 72. The hermetically sealed reactor underwent hydrothermal treatment at 180 °C for 20 h in an oven (Red line RF 115; Thomas Scientific, Swedesboro, NJ, USA). After that, the reaction mixture was cooled to room temperature and thoroughly washed with distilled water to a pH of 7. This was followed by several washing cycles in 0.1 M HCl aqueous solution for removal of Na^+^ ions via protonation and conversion of sodium titanate structure to TiO_2_ nanotubes. The powder was subsequently vacuum dried at 120 °C for 3 h and calcined at 500 °C for 5 h.

### 2.2. Melt Mixing and Injection Molding

The TiO_2_ powder was first dispersed in distilled water followed by the addition of PC granules. Then, the suspension was dried initially by heating on a hot plate, and later, in a vacuum oven at 120 °C. Using a co-rotating, twin-screw extruder (LTE16-40, Lab Tech Engineering Company Ltd., Samut Prakan, Thailand), the dry mixture was twice melt-blended at 290 °C zone temperature and 30 rpm screw speed. The strand was pulled at 8 m/s and pelletized to produce PC/TiO_2_ composite granules as feed material for injection molding.

Prior to injection molding into rectangular plaques, the granules of both as-received PC grade and the PC/TiO_2_ composites were vacuum dried at 120 °C for 4 h for any residual moisture removal. The injection molding process was carried out at 285 °C nozzle temperature for 2 s holding time. The as-received PC grade is designated as P, whereas the composites with 0.5 nanoparticles, 0.1 nanowires, and 0.05 nanosheets and nanotubes (hybrid nanostructures), by weight percent, are referred to as PC/TiO_2_-0.50NP, PC/TiO_2_-0.10NW, and PC/TiO_2_-0.05HN, respectively.

### 2.3. Microstructure and Chemical Characterization

The as-received PC grade and different PC/TiO_2_ composites were characterized for thermal properties using thermogravimetric analysis (TGA/DTA) and differential scanning calorimetry (DSC) techniques. Thermographs of the as-received PC and different PC/TiO_2_ composites were obtained by operating TGA (TA Instruments Pyris 1 diamond Q5000IR, New Castle, WA, USA) from 25 to 700 °C at a rate of 10 °C/min. Thermal properties including glass transition temperature (T_g_), onset of melting, and endothermic peak width were measured under nitrogen flow using a DSC instrument (Mettler Toledo DSC 823e, Columbus, OH, USA) in the temperature range from 25 to 600 °C at 10 °C/min.

TEM microstructure examination of the TiO_2_ nanostructures was carried out through dispersion in ethanol via ultrasonic vibration and probe sonication treatments. A few droplets were poured over the copper holey grid and examined under a field-emission transmission electron microscope (FE-TEM) (JEOL JEM-2100F, Japan) with 200 kV accelerating voltage, LaB_6_ electron gun, and a sub-nanometer resolution of 0.19 nm in ultra-high resolution configuration. For the SEM analysis of the fracture surfaces, a field-emission scanning electron microscope (FE-SEM) (JEOL, JSM7600F) was operated at 5 kV with 4.5 mm working distance for good image resolution. The energy dispersive spectrum analysis of the samples was conducted using an EDS detector (Oxford Instruments, X-act).

Fourier transform infrared spectroscopy (FTIR) of the as-received PC grade was carried out on an attenuation total reflection-Fourier transform infrared spectrometer (JASCO ATR-FTIR-4100, Tokyo, Japan). The room temperature Raman spectra of the TiO_2_ nanotubes and the PC/TiO_2_ composite were recorded with a Jobin Yvon micro Raman spectrometer (T64000, Horiba Scientific, UK) that was operated at 200 mW power using a 514 nm (green laser) to excite the samples.

### 2.4. Environmental Stress Cracking Testing

The ESC tests were performed at constant strain in a 3-point bend test configuration. The injection molded plaques were sectioned as 30.1 × 11 × 2.1 mm^3^ test coupons with fine polished edges. Afterwards, while the test specimen was held at its two ends, a stress of desirable magnitude was introduced at the center by the placement of a pin with a certain diameter to induce sample deflection (*δ*). For a specimen with length, width, and thickness of *L*, *w,* and *t*, respectively, the moment of inertia (*I*) may be computed using the expression:*I = wt^3^/12*(1)

The specimen deflection and the resulting force are correlated in terms of moment of inertia and modulus of elasticity (*E*) as:*δ = FL^3^/48EI*(2)

The applied stress can be calculated using the equation:*σ = (3/2)FL/wt^2^*(3)

Once the test specimen was pre-strained under a stress of desirable magnitude, a thin layer of DOP oil was applied on the top of the specimen using a cotton swab. Then, the specimen was closely examined for any crack initiation and propagation. The time-to-failure which is the time required for surface crack to extend throughout the test specimen width was recorded for different values of applied stress at 40 °C. The results were compared with that of the as-received polycarbonate (P) composite to assess any improvement in ESC resistance. After the tests, the surfaces and the cross-sections of the test specimens were thoroughly examined under SEM.

## 3. Results and Discussion

### 3.1. Characterization of Initial Materials

#### 3.1.1. As-Received Polycarbonate Granules

The electron microscopy of the as-received PC (P) composite revealed the presence of submicrometer-sized particles within the polymer matrix. As shown in Figure 2, the surface microstructure indicated distribution of particles in two different size ranges. While an overall area scan during EDS analysis suggested the presence of C, Ti, Ca, and O in the polycarbonate matrix (inset of Figure 2a), the spot scans over a large and relatively smaller particle (not shown here) indicated their composition to be predominantly Ca/C/O and Ti/C/O, respectively. The added nanoparticles were examined in greater detail by obtaining the residue after thermal decomposition either in a box furnace or after TGA/DTA analysis of the polymer. From SEM examination, the submicron size spherical inorganic nanoparticles with an estimated average size of 385 nm were determined to be CaCO_3_, whereas relatively smaller nanoparticles with an average size of 123 nm were found to be TiO_2_ (Figure 2b).

The FTIR analysis of the as-received polymer granules also confirmed the presence of TiO_2_ and CaCO_3_ nanoparticles (Figure 2c). For the polycarbonate matrix, the positions and bonding assignments of the intense absorption bands are: 731.0 cm^−1^ (C–H out-of-plane bending); 767.6 cm^−1^ (O–C(O)–O stretching mode vibrations); 829.3 cm^−1^ (aromatic C–H out-of-phase vibrations); 1014.5 cm^−1^ (O–C–O stretching mode vibrations); 1080.1 cm^−1^ (C–C–C stretching mode vibrations), 1159.1 cm^−1^ and 1188.1 cm^−1^ (C–O–C stretching vibrations of both alkylene and arylenediphenyl carbonates); 1220.9 cm^−1^ (C=O, C–O isopropylidene vibrations); 1386.7 cm^−1^ (asymmetric and symmetric HC–CH_3_); 1409.9, 1456.2, and 1465.8 cm^−1^ (CH_3_ deformation); 1504.4 cm^−1^ (skeletal vibrations of the aromatic ring), 1602.7 cm^−1^ (ring stretching of phenyl compounds); and 1770.5 cm^−1^ (C=O stretching mode vibrations of the carbonate linkage) [17,18,19,20]. In addition, absorption band characteristics of calcite (CaCO_3_) were noticed to be present at 705.9 and 887.2 cm^−1^, which represented doubly degenerate planar mode (ν_4_, in-plane bending) and out of bending plane (ν_2_ mode) in CaCO_3_, respectively [21]. The presence of TiO_2_ is confirmed through the presence of absorption band at 1108 cm^−1^ that is characteristic of Ti–OH stretching mode vibrations.

#### 3.1.2. TiO_2_ Nanostructures

The sizes and morphologies of the TiO_2_ nanoparticles and nanowires were investigated using SEM and TEM, as presented in Figure 3. From high magnification TEM examination, the TiO_2_ nanoparticles appeared to be of spherical shape with an average size of ~28 nm. Owing to their high surface area, the nanoparticles have a tendency to form agglomerates a few micrometers in size (Figure 3a). The observation of individual nanoparticles in addition to agglomerates as well as the clear identification of individual nanoparticles making up a cluster suggest that these are soft agglomerates and can be de-agglomerated through efficient dispersion treatments and/or use of an appropriate dispersant. Figure 3b presents the TEM microstructure of the TiO_2_ nanowires. The nanowires were polycrystalline with average length and outer diameter values of 7 μm and 150 nm, respectively, thus, revealing an aspect ratio (length-to-diameter ratio) of 47.

The morphology of the TiO_2_ nanostructures obtained from the hydrothermal synthesis are shown by high magnification SEM and TEM images in Figure 4. While TiO_2_ nanotubes were seen to be predominantly formed under the stated conditions of hydrothermal process and afterwards annealing treatment, very thin nanosheets were also noticed to have been formed, as indicated by the white arrows in Figure 4a and the black arrows in the inset of Figure 4b. The nanosheets appear to be very thin and seem to have undergone rolling at the edges to form tubular structure. One such evidence is highlighted by the rounded rectangle in the inset of Figure 4b, where an individual nanosheet terminates as a nanotube as its surface area decreases. In the results presented in Figure 4, the tubular morphology of the nanotubes is evident from the hollow core. The nanotube diameters were ~15–20 nm and the nanotube lengths were of the order of a few hundred nanometers. It has been reported that under hydrothermal conditions, the TiO_2_ precursor nanoparticles dissolve into the concentrated alkaline solution with breaking of the Ti–O–Ti bonds into Ti–O–Na and Ti–OH intermediates via rearrangement followed by formation and growth of sodium titanates lamellar sheets. These nanosheets exfoliate and subsequently grow with a tendency of nanotube formation through curling/scrolling promoted by surface charge variation and/or surface energy minimization [22,23,24,25,26].

### 3.2. PC/TiO_2_ Nanocomposites

#### 3.2.1. Chemical and Thermal Analyses

The Raman spectra of the TiO_2_ HN after hydrothermal synthesis and the PC/TiO_2_-0.1NW composite are shown in Figure 5. For the annealed TiO_2_ nanotubes, the crystal structure is that of anatase phase with Raman active vibration bands positioned at 144.8, 197.4, 398.5, 513.6, and 639.1 cm^−1^. The strong, intense band and the weak shoulder band can be assigned to the Ti–Ti bonds in the octahedral chains, whereas the very weak band at 250.7 cm^−1^ is characteristic of the O–O covalent interactions. The broad bands with medium intensity at 398.5, 513.6, and 639.1 cm^−1^ represent O–Ti–O bending mode vibrations due to moderately distorted TiO_6_^8-^ octahedra in the anatase phase [27,28,29]. In the case of PC/TiO_2_-0.1NW, several Raman bands were observed whose respective positions and the assigned bonding vibration modes are listed in Table 1 [30,31,32]. In addition, Raman active modes representative of the TiO_2_ nanostructures are present. The findings confirm uniform dispersion of the TiO_2_ with a predominantly rutile structure and, to a lesser extent, the anatase phase in the polycarbonate matrix in addition to the presence of CaCO_3_ as calcite [33,34].

The type and weight fraction of the nanoscale additives into a polymer matrix either promote crystallization by offering potential sites for heterogeneous nucleation or inhibit the degree of crystallization by acting as physical obstacles to impose a hindrance to polymer motion, as reported earlier [35,36,37,38]. Investigations into thermal stability of the as-received PC and different PC/TiO_2_ composite formulations were performed through TG/DTA and DSC studies, as demonstrated in Figure 6. From the TG data presented in Figure 6a, it was found that upon heating at 10 °C/min, all the specimens underwent degradation in a single stage over the temperature range of 440–550 °C. For the neat PC, the rate of weight loss became maximum at 523.8 °C. The three nanocomposite formulations exhibited distinctly different behaviors in terms of onset of thermal decomposition. While the addition of 0.5 wt.% TiO_2_ nanoparticles (PC/TiO_2_-0.5NP composite) did not affect the thermal properties of the resulting nanocomposites, incorporation of as little as 0.1 wt.% of the TiO_2_ nanowires (PC/TiO_2_-0.1NW) caused significant deterioration in thermal stability of the resulting nanocomposites. A comparison of the data for the as-received PC and PC/TiO_2_-0.1NW composite revealed that for the same weight loss *x*, the corresponding temperature decreased by up to 25 °C in the case of PC/TiO_2_-0.1NW. In addition, the PC/TiO_2_-0.05HN that contained only 0.05 wt.% TiO_2_ hybrid nanostructures showed improvement in thermal stability over the as-received PC grade.

The main thermal degradation pathways in PC are chain scission and hydrolysis/alcoholysis with subsequent evolution of mostly alkyl-substituted phenol structures during the initial degradation stage and that of less aliphatic substituents at the later stage. Furthermore, H_2_O, CO, CO_2_, CH_4_, and other lower molecular weight hydrocarbons including aldehydes, carbonyls, and ketones are also released [19,39]. In the absence of any surface pretreatment of the added nanostructures, the difference in thermal behavior can be attributed to the morphology and relative content of each TiO_2_ polymorph. Although there was no noticeable change in the T_g_ value as compared with that of the as-received PC grade (143.5 °C) (Figure 6b), a change in thermal properties was usually associated with the free volume and the molecular mobility upon heating and/or degradation. The DSC thermograms exhibited T_g_ values upon initial heating to be ~140 °C followed by varying degrees of process ordering, as indicated by the shallow dip in the curve at temperatures in the range from 220 to 300 °C, particularly for the PC/TiO_2_-0.5NP composite. Another reason for the latter observation is likely to be the smaller sample volume. In the case of composite samples, the decrease in the intensity of the T_g_, as manifested by the step size, revealed an increase in the crystallinity of the polymer matrix upon nanoscale reinforcement addition. The strong endothermic peak is characteristic of the polymer melting behavior, whereas the presence of a small exothermic peak just before that (in the case of PC/TiO_2_-0.5NP and PC/TiO_2_-0.1NW samples) may be due to polymer relaxation and/or crystallization. The sample size determines both data sensitivity and reproducibility with a large sample size yielding increased sensitivity and decreased resolution. The respective values of temperatures representative of the onset of melting, endothermic peak, and peak width are listed in Table 2. In contrast with PC/TiO_2_-0.5NP and PC/TiO_2_-0.1NW, the addition of 0.05 wt.% TiO_2_ NH (both nanosheets and nanotubes) induced a slight improvement in thermal stability, albeit with a significant delay in onset of thermal decomposition (by ~21 °C). Recent studies on PC-based nanocomposites have also reported enhancement in thermal stability upon incorporation of fused silica nanoparticles or pristine/modified halloysite nanotubes [20,40]. Once melting is complete, all the composites presumably experience oxidative decomposition and/or vaporization, as evident from the negative slope indicating an endothermic change. The deterioration in thermal properties in PC/TiO_2_-0.1NW may be attributed to several factors including rigid morphology of the added TiO_2_ nanowires, weak PC/TiO_2_ interfacial interaction, and void formation at the PC/TiO_2_ interfacial area.

#### 3.2.2. Environmental Stress Cracking (ESC) Tests

The data from the ESC tests under different temperatures and pre-strain conditions are graphically presented in Figure 7. It was found that as the degree of specimen pre-straining for the as-received PC increased from 15 to 45 MPa, the time-to-failure dropped dramatically from 124 to about 1 h, despite a modest drop in the temperature from 40 to 22 °C. This observation underlines the importance of the extent of pre-straining to induce stress into a test specimen during bending. Upon application of DOP oil to the pre-strained sample surface, the chemically exposed polymer was weakened due to initiation and subsequent growth/propagation of micro crazes, crazes, and cracks. Keeping the dimensions of the specimens identical, the ESC test conditions were chosen such that the test specimens failed over a period of a few hours as it would clearly differentiate between performance attributes of different composite formulations.

PC susceptibility to alkaline environments has been reported in terms of deterioration in thermal properties upon exposure to an alkaline component. One such report was based on melt blending PC with potassium titanate whiskers using a coupling agent such as tetrabutyl orthotitanate for surface modification [41]. Keeping this fact in mind, different TiO_2_-based polymorphs including nanoparticles, nanowires, and nanotubes/nanosheets were investigated. In contrast with the ESC behavior exhibited by the TiO_2_ nanoparticles and nanowires, the addition of TiO_2_ nanotubes and nanosheets significantly improved the ESC resistance of the resulting composite, as indicated by a much greater time-to-failure for this composition.

#### 3.2.3. Microstructural Analysis of PC and PC/TiO_2_ Composites

The surface and cross-section of the PC coupon after the ESC test are shown in Figure 8. Several images were recorded and stitched to generate complete surface and cross-sectional cracking profiles. In the case of sample P, initially only one crack nucleated at the surface upon exposure to DOP oil and as the time progressed, this crack propagated along the surface, while at the same time, more cracks appeared in its vicinity. Figure 8a provides a low magnification SEM microstructure of the test specimen’s surface in which the main crack originated, but very hostile conditions of oil exposure together with high residual stress level caused branching of this crack in addition to the development of more cracks that eventually converged to complete failure. Pre-strain induced by the pin at the opposite surface of the specimen initiated one or multiple surface cracks that acted as a notch with stress concentration at its tip. In addition to growth along the surface, crack propagation also occurred along the specimen’s cross-section through exposure of new area to the DOP oil that in combination with stress concentration deteriorated the polymer matrix causing rupture along the sample thickness. During this process, the microstructure underwent significant changes with different features that were characteristic of specific conditions at a microscopic scale. From an examination of the P sample fracture cross-section (Figure 8b), it was found that a greater degree of specimen pre-straining caused extensive crazing along the cross-sectional area beneath the surface exposed to the DOP oil and also to a significant depth.

While the co-existence of oil and stress promotes extensive crazing in the beginning and even in some areas of the central region of the specimen cross-section, factors such as greater surface area, chemical degradation and/or enhanced degree of solubility in regions of high stress at flaws and crack tips, as well as loss in toughness due to a reduction in molecular weight of the polymer contribute towards a shift from crazing to cleavage fracture [42]. Such transition is manifested in Figure 9a where coarse craze structure formation is noticed to eventually turn into branches or fine crazes at its end. The crazes are noticed to be ~500 µm in length with an inter-craze spacing of 350–400 nm. In addition, fine crazes are also present within the coarse craze structure along with crater-like regions formed presumably due to polymer dissolution by DOP. As the crack further proceeds into the sample thickness, the exposure of more surface to the DOP oil as well as stress concentration cause polymer degradation through dissolution and a change in the glass transition temperature. This is evident in Figure 9b which shows signs of extensive material flow with diminished and deteriorating craze structure. The extensive polymer degradation is noticed at low and high magnification microstructures presented in Figure 9c,d through polymer matrix softening in addition to the generation of surface porosity. Additionally, the submicron-sized particles also appear to be embedded into the polymer matrix, although there are certain particles that seem to have undergone pullout or detachment from the matrix, as indicated by the white arrows in Figure 9d. This fracture morphology shows areas with high densification near the notch that restricts segmental motion within the polymer matrix.

The fracture cross-section of the PC/TiO_2_-0.1NW composite after the ESC test is shown in Figure 10. As compared to PC, the fracture surface at low magnification (Figure 10a) appears to be mostly smooth with rough surface and craze formation only confined to the top-end region of the surface over which DOP was applied. The segmental tearing and rough, faceted surface characterize massive deformation. Such tear texture is indicative of regions that prevent rapid crack growth due to high impact strength and large deformation areas with a subsequently large absorption of energy required for crack propagation [40]. A closer look at the crazes in the near-surface cross-section area (Figure 10b) indicates that they are a few tens of micrometers apart with subsequent branching out along their length and that they undergo deterioration due to subsequent DOP exposure of the region further below the top surface. Figure 10c,d further elaborate upon this observation where polymer flow patterns were seen, presumably due to lowering of the T_g_ as a result of the synergistic effect of DOP oil and high localized stress concentration.

### 3.3. Effect of TiO_2_ Hybrid Nanostructures

The fracture surface of the PC/TiO_2_-0.05HN composite after DOP oil cracking is presented in Figure 11. At low magnification, the microstructural features were, to a certain extent, similar to those seen in other samples such as crazes in the near-surface area with a highly rough area indicative of massive deformation beneath and finally quite a smooth large area with characteristic polymer degradation and flow as the crack progressed further into the specimen cross-section. High magnification views of certain regions, however, reveal some interesting and unique features providing insight into the failure mechanism. The area away from the near-surface region (Figure 11d) is not as smooth as it seemed at a low magnification, rather, it appears quite rough with void formation that is characteristic of high resistance to crack propagation despite polymer degradation upon exposure to DOP oil. Very rough patches representative of massive deformation, when examined at a high magnification, reveal resistance to crack propagation through both bridging and pullout mechanisms, as evident from Figure 11e,f. Quite interestingly, both nanotubes as well as nanosheets seem to bridge the gap between polymer segments as a testament to their high strength and efficient load-bearing capabilities. At the same time, some of the hybrid nanostructures are noticed to have been broken or pulled out of the polymer matrix under highly localized loads. Figure 11f shows a bundle mostly comprised of nanotubes that effectively bridge the gap as the crack progresses. This observation is in sync with an earlier report on PC/CNT composites with 2 weight percent CNT, where a brittle-to-ductile transition was described in terms of reductions in the crack dimensions from 0.6 to 0.09–0.27 μm owing to crack bridging by the CNT with an effective associated enhancement in the resistance to crack propagation [43].

The crack bridging by the TiO_2_ nanosheets and nanotubes, as in the case of PC/TiO_2_-0.05HN is more clearly demonstrated by the SEM microstructures presented in Figure 12. While the crack is wide open at most places, the space narrows down in the region where nanotubes bridge the gap. The difference in crack width between such regions is estimated in Figure 12a, where three TiO_2_ nanotubes appear to have effectively bridged the gap. Incidentally, when the electron beam was kept focused at this area for few minutes, the e-beam–PC interaction caused ruptures in two of the three nanotubes (Figure 12b). Since elastic stretching of the nanotubes was recovered upon breaking, the elastic strain value was computed from measurements of the nanotube segments before and after ruptures and was found to be ~5%.

To further understand the role of TiO_2_ nanostructures towards improvement in ESC and toughness, a small notch in the PC/TiO_2_-0.05HN sample was produced followed by fracture in liquid nitrogen and subsequent examination of the fracture surface under SEM, as presented in Figure 13. At low magnification (Figure 13a), there is strong evidence of bridging between polymer segments by the TiO_2_ nanostructures, as pointed out by the areas inside the white circles/ellipses. This is also testament to their homogeneous dispersion and strong integration with the PC matrix. The fact that these nanostructures are intact with both ends embedded in the PC matrix implies the interfacial shear strength (IFSS) to be very high with efficient load-bearing capabilities of the reinforcing nanostructures having both tubular and sheet morphologies. Generally, the IFSS value is very high when there is covalent bonding between the polymer matrix and the reinforcing phase [44,45]. The area (Figure 13a) is thoroughly viewed at higher magnification (Figure 13b). The efficient load-bearing characteristics of the TiO_2_ nanosheets are demonstrated by the stretching, and in some cases, partial tearing/branching of the nanosheets. The length and width of the stretched nanosheets have values in the order of a few hundred nanometers and <100 nm, respectively. The gradual thinning of the TiO_2_ nanostructure from about 100 nm at one edge to ~40 nm can be seen in Figure 13c. These observations illustrate the anchoring role of the TiO_2_ hybrid nanostructures during crack propagation, while the composite specimen undergoes localized stress concentration during ESC or impact loading.

## 4. Conclusions

TiO_2_ nanostructures with a combination tubular and sheet-like morphology, referred to as hybrid nanostructures (HNs) can be produced using a hydrothermal process. Typical dimensions of TiO_2_ nanotubes and nanosheets were ~15–20 nm and several hundred nanometers, respectively. PC/TiO_2_ composites with TiO_2_ nanoparticles, nanowires, or hybrid nanostructures can be fabricated from aqueous colloidal dispersion of these nanostructures followed by melt extrusion with PC granules. Environmental stress cracking (ESC) tests with certain degrees of pre-straining and DOP oil exposure assessed any improvement in ESC resistance of the PC/TiO_2_ composites as compared to the as-received PC. While the addition of TiO_2_ NP or NW led to meagre improvements in ESC resistance to DOP oil, a significant improvement was seen in the case of TiO_2_ HN upon incorporation into the PC matrix. The time to complete rupture increased by several folds, in one case from ~3.1 to 70 h, for the PC/TiO_2_-0.05HN composite. Owing to their high tensile strength and high surface area, such TiO_2_ nanostructures exhibit highly efficient load bearing characteristics despite chemical degradation of the PC matrix. Qualitatively, the localized interaction of concentrated stress and/or propagating microcracks with TiO_2_ HN within the PC matrix was more intense as well as highly effective, as further confirmed from microstructure examination of a notched, cryogenically fractured specimen.

## Figures and Tables

**Figure 1 polymers-14-03693-f001:**
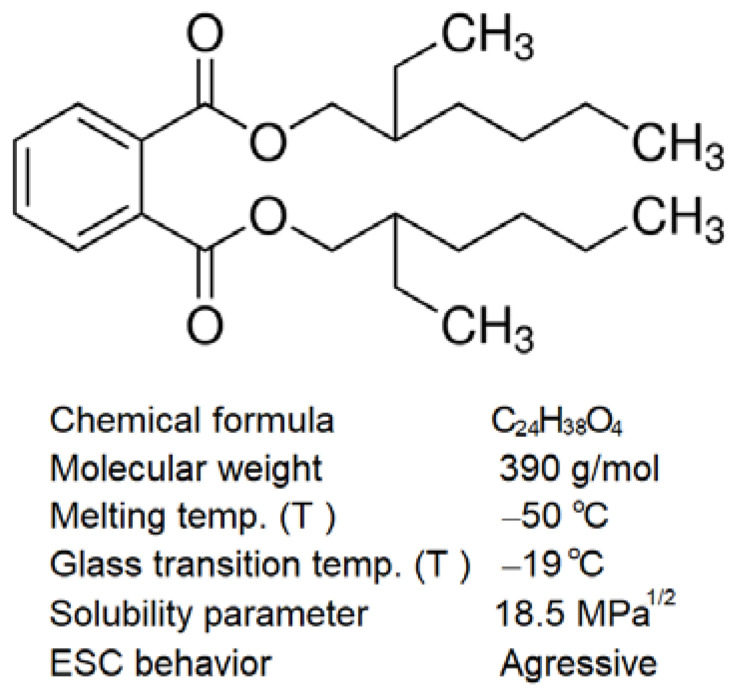
Molecular structure and properties of dioctyl phthalate (DOP) oil.

**Figure 2 polymers-14-03693-f002:**
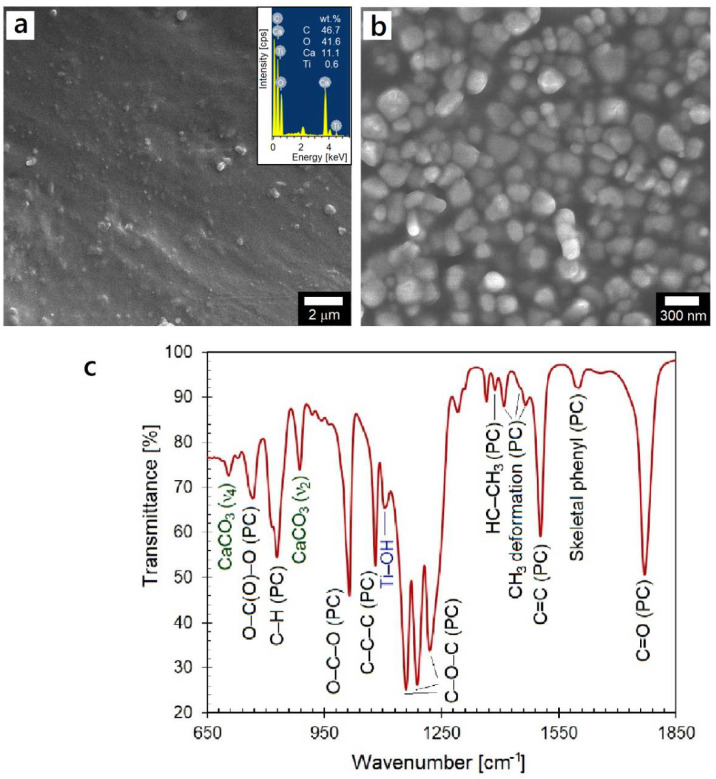
(**a**,**b**) SEM micrographs of the as-received PC granules with EDS data (inset) and polymer residue upon thermal decomposition; (**c**) FTIR spectrum indicating bonding characteristics of the PC matrix and the nanoparticles.

**Figure 3 polymers-14-03693-f003:**
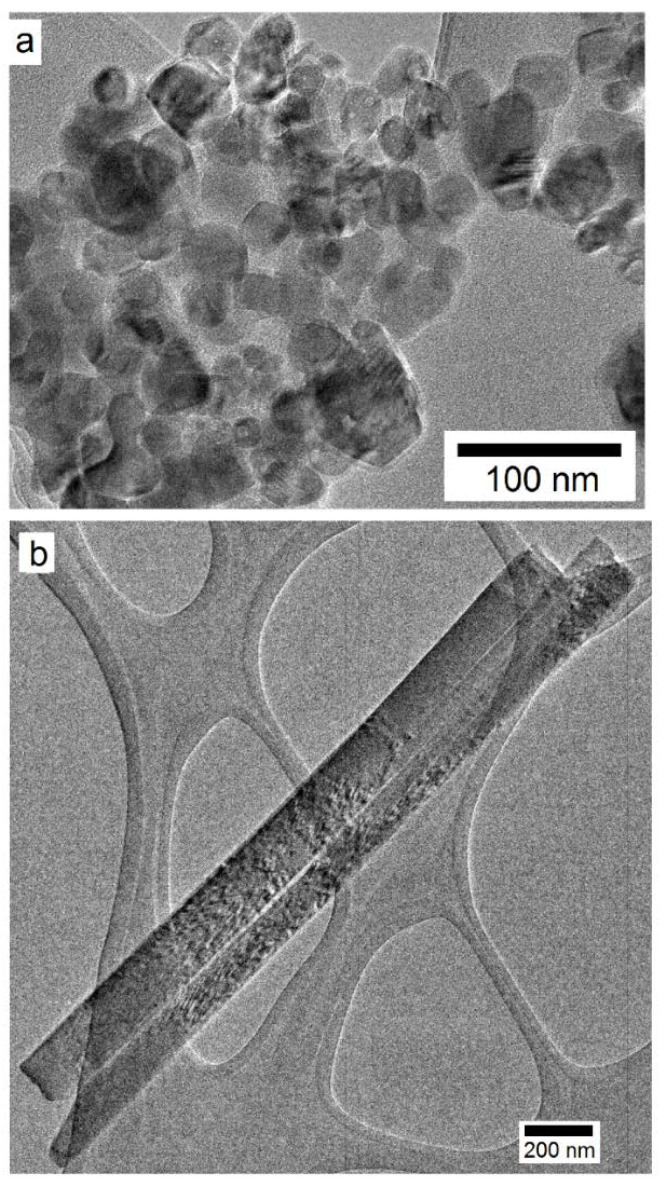
TEM microstructures of: (**a**) TiO_2_ nanoparticles (NP); (**b**) TiO_2_ nanowires (NW), used as nanoscale filler in the as-received PC grade.

**Figure 4 polymers-14-03693-f004:**
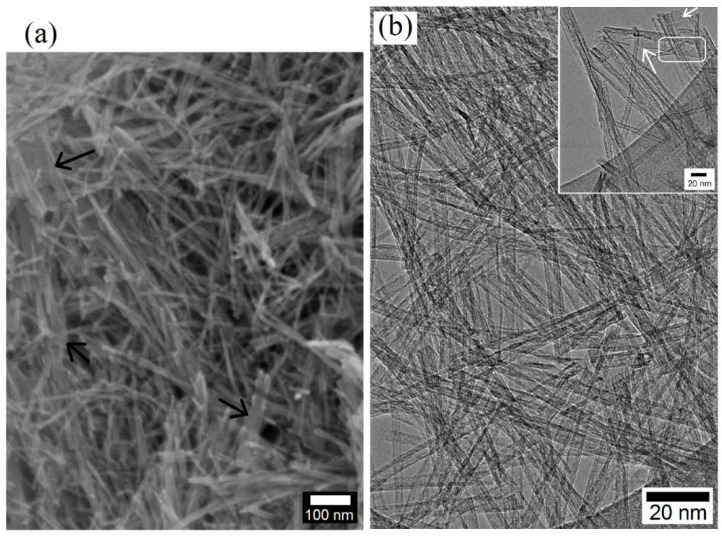
High magnification views of TiO_2_ hybrid nanostructures (HN) produced via a hydrothermal process, indicating both sheet-like and tubular morphologies: (**a**) SEM micrograph and (**b**) TEM microstructures with inset showing nanotubes (white arrows) and nanosheet (enclosed area).

**Figure 5 polymers-14-03693-f005:**
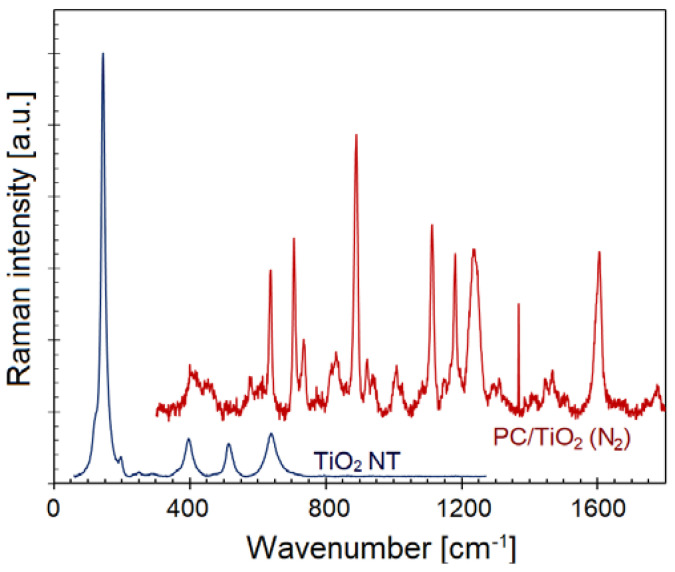
Raman spectra of the TiO_2_ HN powder hydrothermal synthesis and PC/TiO_2_-NW0.10 composite.

**Figure 6 polymers-14-03693-f006:**
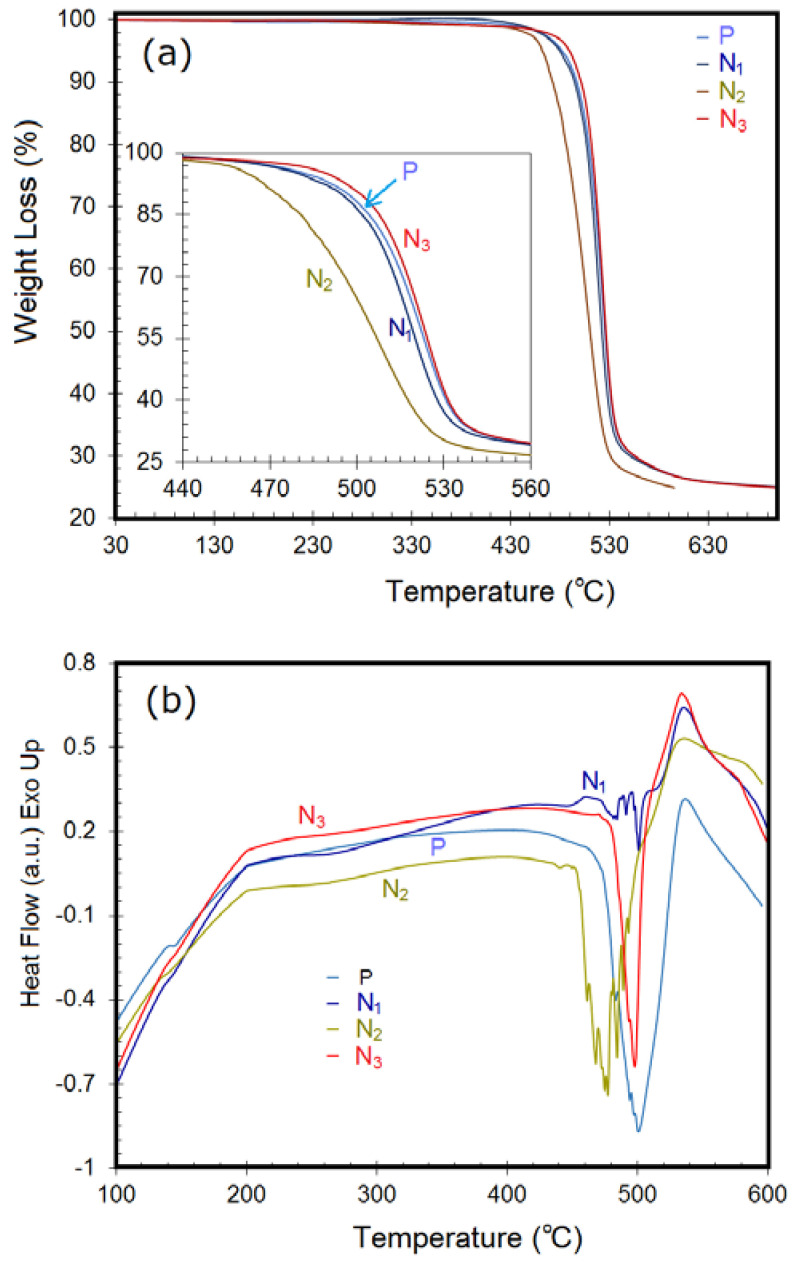
Thermographic analysis of PC and different PC/TiO_2_ composites: (**a**) Overall thermogravimetric curves with inset differentiating the onset of thermal decomposition; (**b**) DSC thermograms indicating T_g_ values. The samples P, N_1_, N_2,_ and N_3_ represent PC, PC/TiO_2_-0.5NP, PC/TiO_2_-0.1NW, and PC/TiO_2_-0.05HN, respectively.

**Figure 7 polymers-14-03693-f007:**
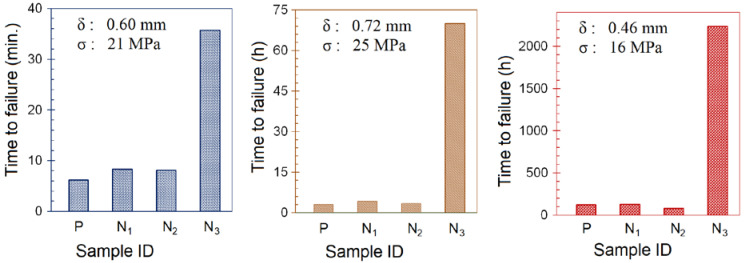
Bar charts showing time-to-fracture for PC and PC/TiO_2_ composites after ESC tests in DOP oil under different temperatures and pre-strain values.

**Figure 8 polymers-14-03693-f008:**
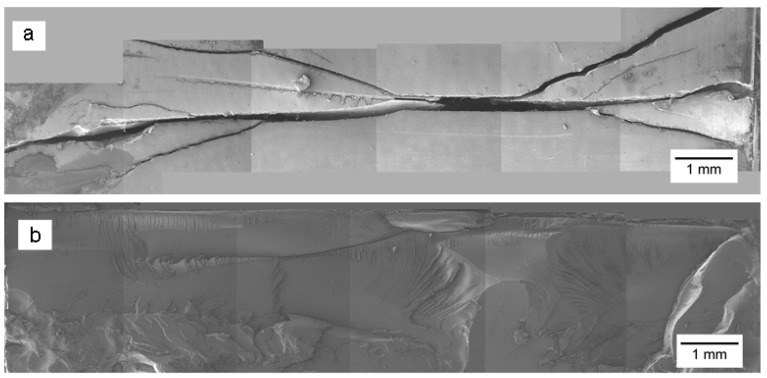
Stitched SEM microstructure of the PC after the ESC test at high pre-strain and in the presence of DOP oil: (**a**) Surface crack profile; (**b**) fractured cross-section.

**Figure 9 polymers-14-03693-f009:**
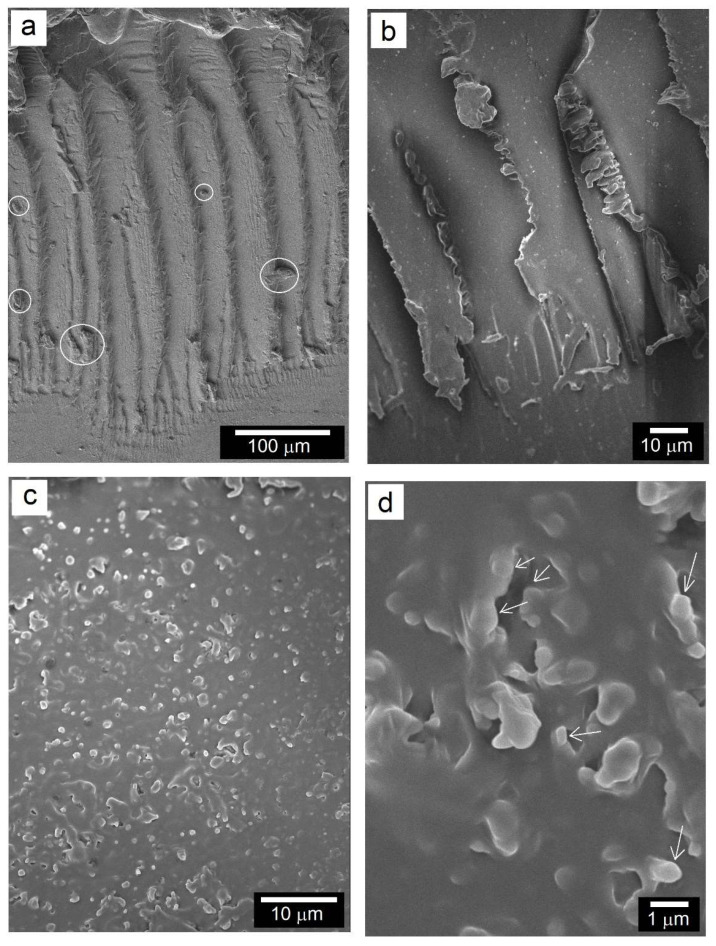
Low and high magnification views of different fracture features on PC sample cross-section after DOP exposure under high constant strain: (**a**,**b**) craze structure and its termination and/or branching, and (**c**,**d**) polymer dissolution leading to crater formation. The circles and arrows indicate craze branching and filler particle detachment, respectively.

**Figure 10 polymers-14-03693-f010:**
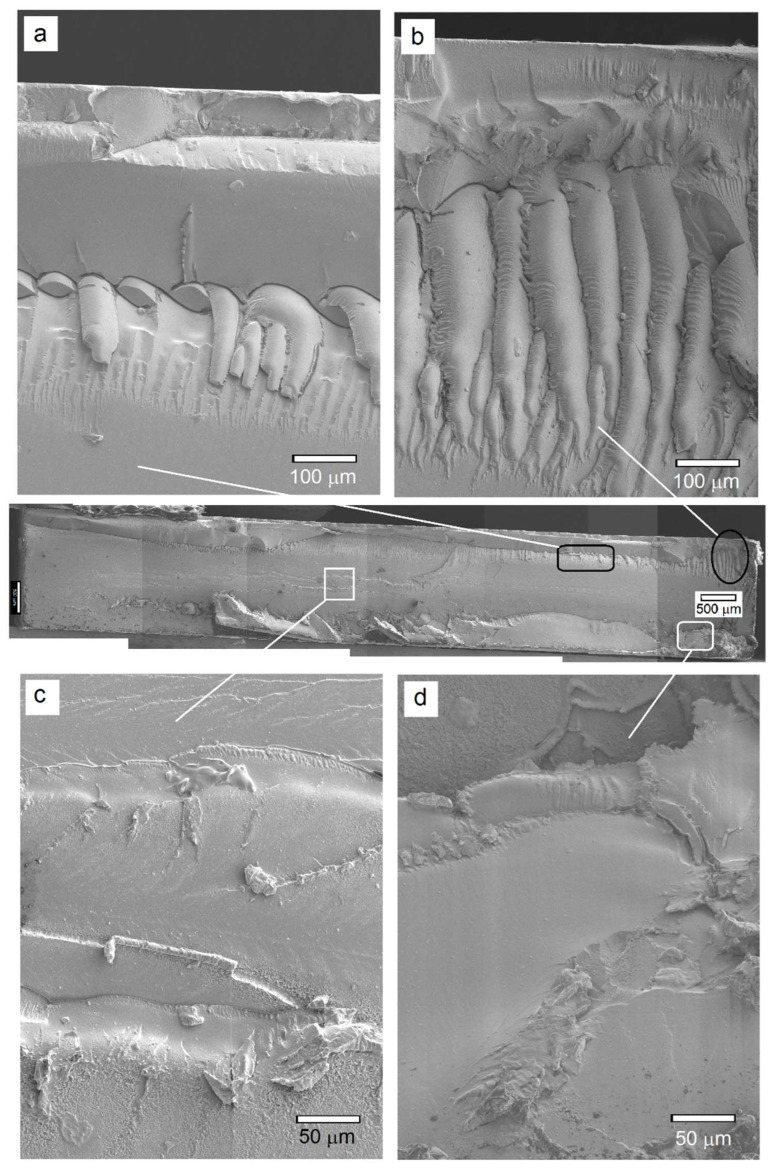
(Center) SEM cross-sectional view of the fractured PC/TiO_2_-0.5NP composite: (**a**,**b**) Near-surface regions indicating crazing and matrix degradation; (**c**) large, relatively smooth region in the mid-section demonstrating smooth cleavage failure due to matrix softening and flow; (**d**) rough, grainy appearance near the back surface of test specimen.

**Figure 11 polymers-14-03693-f011:**
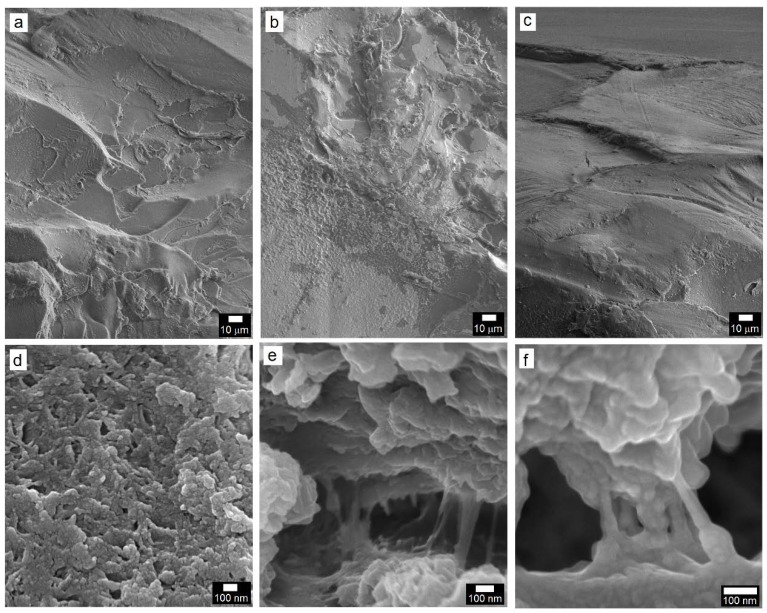
SEM microstructures of the fractured PC/TiO_2_-0.5HN composite: (**a**,**b**) Near-surface region revealing craze formation and matrix degradation during the DOP test; (**c**) highly smooth region polymer matrix degradation and flow in the presence of DOP and localized stress; (**c**) large, smooth region near the back surface of the test specimen; (**d**) rough, voided region characteristic of high resistance to crack propagation; (**e**,**f**) within crack segments showing crack bridging by the TiO_2_ nanosheets and nanotubes.

**Figure 12 polymers-14-03693-f012:**
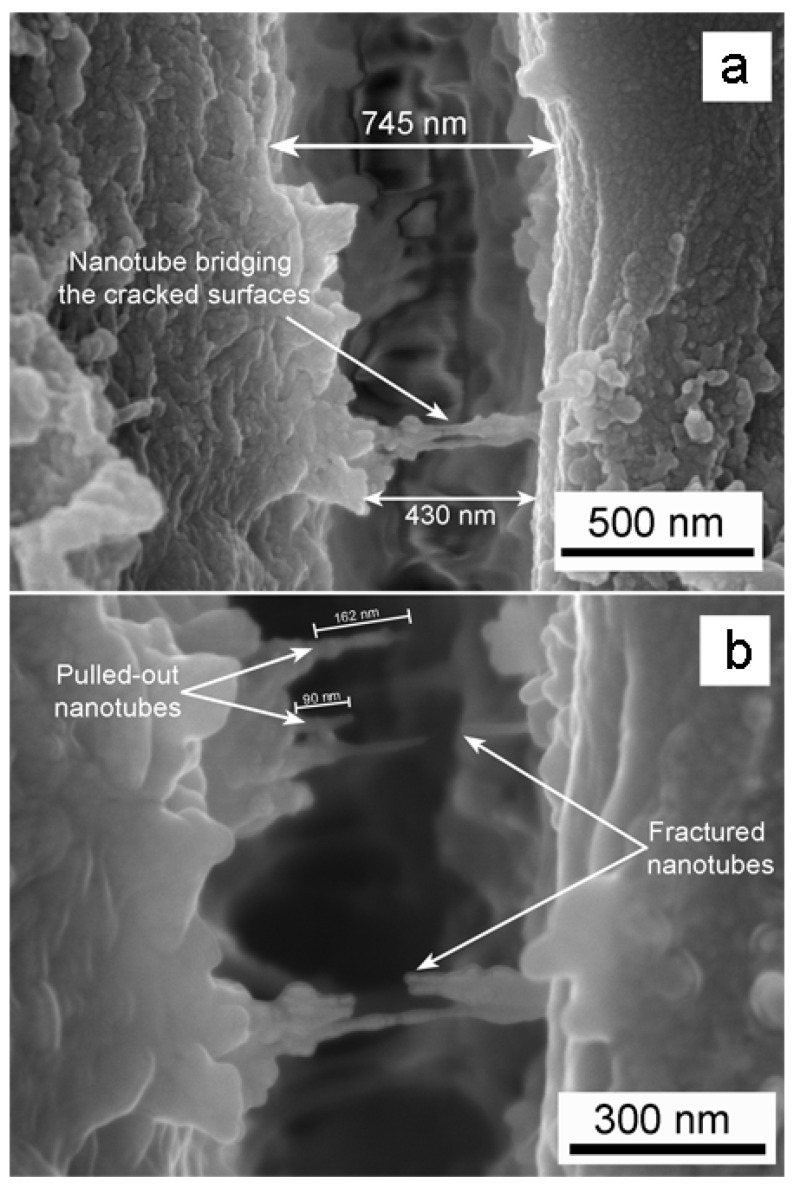
PC/TiO_2_-0.05HN fractography at high magnification: (**a**) A crack opening revealing crack bridging by the TiO_2_ nanosheets and nanotubes; (**b**) TiO_2_ NT pullout during electron beam scanning of a small area inside a crack.

**Figure 13 polymers-14-03693-f013:**
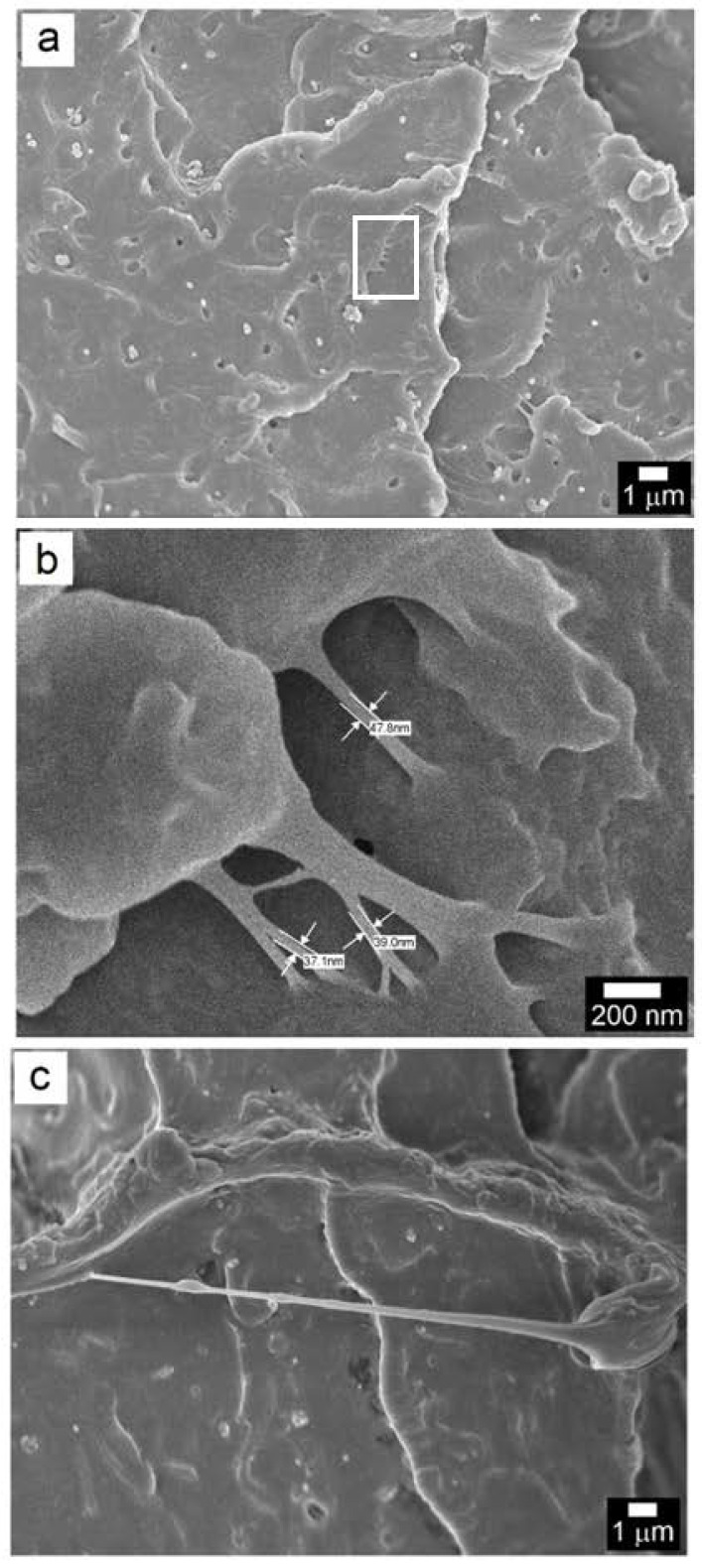
PC/TiO_2_-0.05HN cross-section after liquid nitrogen fracture of the notched composite: (**a**,**b**) Low and high magnification micrographs, and (**c**) region representing gradual thinning of bridge formed by TiO_2_ nanosheets and nanotubes between polycarbonate segments.

**Table 1 polymers-14-03693-t001:** Assignment of Raman modes in pure TiO_2_ powder and PC/TiO_2_-0.10NW composite.

Intensity	Frequency (cm^−1^)	Phase	Vibration Mode
vs	145	Anatase TiO_2_	E_g_ mode
sh	197	Anatase TiO_2_	B_1g_ mode
m	399	Anatase TiO_2_	A_1g_ mode
m	514	Anatase TiO_2_	B_1g_ mode
m	639	Anatase TiO_2_	E_g_ mode
w	400	Anatase TiO_2_/PC	A_1g_ mode; O–C–O bend
w	453	Rutile TiO_2_	E_g_ mode
w	581	Polycarbonate	Phenyl ring vibration
vw	614	Rutile TiO_2_	
s	638	Anatase TiO_2_/PC	E_g_ mode; Phenyl ring def. (i.p.)
s	706	Polycarbonate	Phenyl ring def. (o.p.)
m	735	Polycarbonate	C–H bend (o.p.)
sh	817	Polycarbonate	CH wag (o.p.)
w	830	Rutile TiO_2_	B_2g_ mode
sh	837	Polycarbonate	Phenyl ring vibration
vs	890	Polycarbonate	C–CH_3_ stretch
w	897	Polycarbonate	O–C(O)–O stretch
w	940	Polycarbonate	CH wag (o.p.)
w	1008	Polycarbonate	Ring stretch
sh	1025	Polycarbonate	C–C stretching
w, sh	1084	Calcite (CaCO_3_)	υ_1_ Symmetric stretch (CO_3_)^2–^
s	1112	Polycarbonate	CH wag (i.p.)/C–O–C stretch
s	1180	Polycarbonate	CH wag
s	1238	Polycarbonate	C–O stretch
s, sh	1245	Polycarbonate	C–O–C stretch
w	1297	Polycarbonate	C–O–C stretch
m	1368	Polycarbonate	CH_3_ bend
w	1446	Polycarbonate	CH_3_ symmetric bend
w	1468	Polycarbonate	CH_3_ asymmetric bend
s	1605	Polycarbonate	Phenyl ring stretch
w	1780	Polycarbonate	C=O stretch

s, strong; m, medium; w, weak; vw, very weak; sh, shoulder; i.p., in-plane; o.p., out-of-plane.

**Table 2 polymers-14-03693-t002:** Temperatures corresponding to different weight losses from TGA results and DSC data for the as-received PC and different PC/TiO_2_ composites.

ID	Temp. for Different Weight Loss (°C)						Melting Peak (°C)			
	T_−5_	T_−10_	T_−20_	T_−30_	T_−50_	T_−70_	Onset	Peak	End	Width
PC	482.1	497.0	509.2	515.8	525.5	555.2	456.1	500.6	532.0	75.9
PC/TiO_2_-0.5NP	479.9	494.8	506.9	513.1	522.5	551.3	459.9	501.1	505.7	45.8
PC/TiO_2_-0.1NW	462.0	472.5	486.2	495.9	510.4	531.1	450.3	477.3	498.8	48.5
PC/TiO_2_-0.05HN	489.7	501.4	511.8	517.7	526.5	555.8	477.0	497.8	508.1	31.1

## Data Availability

Not applicable.

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
