# Peer review of "Polycarbonate/Titania Composites Incorporating TiO2 with Different Nanoscale Morphologies for Enhanced Environmental Stress Cracking Resistance in Dioctyl Phthalate"

_polymers, 2022, doi:10.3390/polym14173693_

Round 1

Reviewer 1 Report

The authors have fabricated "Polycarbonate/Titania Composites" for Enhanced Environment Stress Cracking Resistance in Dioctyl Phthalate. The work is deeply conducted and can be published after incorporating the suggestions.

Abstract: The abstract needs to be more quantitative with the size of TiO2 and the stress cracking values.

Introduction: The introduction discusses ESC and PC primarily but lacks an introduction to the titania and the reason for adding to form a better composite.
Titania is a highly renowned material with phases, i.e., Anatase and Rutile, and is used for various applications from energy conversion to storage or antimicrobial activity. Here are some references which can be added further to introduce Titania and its various application:

1) Khanna, Sakshum, et al. "Fabrication of graphene/Titania nanograss composite on shape memory alloy as photoanodes for photoelectrochemical studies: Role of the graphene." International Journal of Hydrogen Energy (2022).

2) Gohil, Manisha, and Girish Joshi. "Perspective of polycarbonate composites and blends properties, applications, and future development: A review." Green Sustainable Process for Chemical and Environmental Engineering and Science (2022): 393-424.

3)Khanna, Sakshum, et al. "Investigation of Thermophysical Properties of Synthesized N-Hexacosane-Encapsulated Titania Phase Change Material for Enhanced Thermal Storage Application." Recent Advances in Mechanical Infrastructure. Springer, Singapore, 2022. 107-118.

Result: The result has been deeply written, but the authors need to add X-ray diffraction for the individual material and the composite in order to compare the change in the crystallinity of the composite.
Also, why is there a significant shift in the DSC curve for N1 sample during the cooling cycle? Please add for more clarity.

Conclusion: The conclusion is well written, but modifying it quantitatively will increase the clarity to the reader. 

There are some grammatical errors in the manuscript. Kindly modify them in the revised version.

Author Response

The responses made against each comment are listed in the uploaded document.

Reviewer 2 Report

In this manuscript, the authors investigated the effect of TiO2 nanoparticle morphology on the thermal properties and ESC resistance of the PC/TiO2 composites in DOP oil. The manuscript is well written, I have some minor suggestions.

1. I suggest the authors to add some background information about TiO2 nanoparticles in the introduction section. Why TiO2 was chosen in this study? What are the common crystal structures of TiO2? What are the commonly used methods to synthesize TiO2 nanoparticles with different shapes? Some papers can be cited: "High pressure induced atomic and mesoscale phase behaviors of one-dimensional TiO2 anatase nanocrystals." MRS Bulletin (2022): 1-6.; "Synthesis of nanostructured reduced titanium oxide: crystal structure transformation maintaining nanomorphology." Angewandte Chemie 123.32 (2011): 7556-7559.;  "Corrosion protection of 316 L stainless steel by a TiO2 nanoparticle coating prepared by sol–gel method." Thin Solid Films 489.1-2 (2005): 130-136.

2. Detailed experiment conditions for DSC and TGA such as temperature range and heating rate should be given in the experimental section. 

3. What is P0 is Figure 7?

Author Response

Kindly see the uploaded document for all the responses made against the reviewer's comments.

Round 2

Reviewer 1 Report

The author has improved certain areas, but still, the literature section has not been added with suggested articles or current literature. Thus the work can be accepted after minor revision.

As per the author's response, "The suggested research articles have been added to the list of references consulted for this work" the updated references and literature was missing in the revised version of the manuscript. Kindly incorporate the recent or suggested literature in the work.